# Oxidative Stress in Postmenopausal Women with or without Obesity

**DOI:** 10.3390/cells12081137

**Published:** 2023-04-12

**Authors:** Giulia Leanza, Caterina Conte, Francesca Cannata, Camilla Isgrò, Alessandra Piccoli, Rocky Strollo, Carlo Cosimo Quattrocchi, Rocco Papalia, Vincenzo Denaro, Mauro Maccarrone, Nicola Napoli, Anna Maria Sardanelli

**Affiliations:** 1Department of Medicine and Surgery, Unit of Endocrinology and Diabetes, Campus Bio-Medico University of Rome, 00128 Rome, Italy; 2Department of Human Sciences and Promotion of the Quality of Life, San Raffaele Roma Open University, 00166 Rome, Italy; 3Department of Endocrinology, Nutrition and Metabolic Diseases, Istituto di Ricovero e Cura a Carattere Scientifico (IRCCS) MultiMedica, 20900 Milan, Italy; 4Department of Translational Biomedicine and Neuroscience ‘DiBraiN’, University of Bari “Aldo Moro”, Pi-azza G. Cesare 11, 70124 Bari, Italy; 5Department of Science and Technology for Sustainable Environment and One Health, Campus Bio-Medico University of Rome, Via Alvaro del Portillo 21, 00128 Rome, Italy; 6Department of Medicine, Unit of Diagnostic Imaging and Interventional Medicine, Fondazione Policlinico Universitario Campus Bio-Medico, 00128 Rome, Italy; 7Department of Medicine, Unit of Orthopaedic and Trauma Surgery, Fondazione Policlinico Universitario Campus Bio-Medico, 00128 Rome, Italy; 8Department of Biotechnological and Applied Clinical Sciences, University of L’Aquila, Via Vetoio snc, 67100 L’Aquila, Italy; 9European Center for Brain Research, Santa Lucia Foundation IRCCS, 00164 Rome, Italy; 10Department of Medicine and Surgery, Unit of Biochemistry and Molecular Biology, Campus Bio-Medico University of Rome, 00128 Roma, Italy

**Keywords:** obesity, oxidative stress, menopause, visceral fat, lipid peroxidation, oxidative damage

## Abstract

Oxidative stress, a key mediator of cardiovascular disease, metabolic alterations, and cancer, is independently associated with menopause and obesity. Yet, among postmenopausal women, the correlation between obesity and oxidative stress is poorly examined. Thus, in this study, we compared oxidative stress states in postmenopausal women with or without obesity. Body composition was assessed via DXA, while lipid peroxidation and total hydroperoxides were measured in patient’s serum samples via thiobarbituric-acid-reactive substances (TBARS) and derivate-reactive oxygen metabolites (d-ROMs) assays, respectively. Accordingly, 31 postmenopausal women were enrolled: 12 with obesity and 19 of normal weight (mean (SD) age 71.0 (5.7) years). Doubled levels of serum markers of oxidative stress were observed in women with obesity in women with obesity compared to those of normal weight (H_2_O_2_: 32.35 (7.3) vs. 18.80 (3.4) mg H_2_O*_2_*/dL; malondialdehyde (MDA): 429.6 (138.1) vs. 155.9 (82.4) mM in women with or without obesity, respectively; *p* < 0.0001 for both). Correlation analysis showed that both markers of oxidative stress increased with an increasing body mass index (BMI), visceral fat mass, and trunk fat percentage, but not with fasting glucose levels. In conclusion, obesity and visceral fat are associated with a greater increase in oxidative stress in postmenopausal women, possibly increasing cardiometabolic and cancer risks.

## 1. Introduction

Menopause is a condition precipitated by the permanent cessation of ovarian follicular function, leading to a decline in estrogen levels combined with an enhanced risk of cardiovascular disease (CVD), including arterial hypertension, coronary artery disease (CAD), and cerebrovascular disease [1]. Menopause is also associated with weight gain and worsening body composition with a shift from gynoid to android fat distribution [2]. Although the responsible mechanisms have not been completely clarified, changes in energy homeostasis due to aging; estrogen deficiency; sleep disturbances; changes in adipokines; gut hormones; and gut microbiota seem to play a role in weight gain and the worsening of body composition during menopause [3]. It has been estimated that approximately 70% of women of perimenopausal age are overweight or obese, both of which are associated with excess morbidity and mortality [4]. Obesity may have detrimental effects on a postmenopausal woman’s health, which are related to an increased risk of cancer [5], particularly breast cancer [6,7], and an increased risk of CVD [8]. Obesity is also associated with increased oxidative stress [9], a key mediator of CVD. In fact, the generation of reactive oxygen/nitrogen species (ROS/RNS) such as superoxide (O_2_^−^), hydrogen peroxide (H_2_O_2_), and peroxynitrite (ONOO^−^) is promoted by an imbalance between antioxidant and oxidant factors [10,11]. These factors have the potential to induce the oxidative modification of major cellular macromolecules (i.e., proteins, DNA, and lipids), thereby altering subcellular organelles and causing endothelial dysfunctions that promote atherosclerosis, which are fundamental events in the development of CVD [12]. Polyunsaturated fatty acids are particularly vulnerable to ROS, to which they yield primary products of lipid peroxidation, i.e., lipid hydroperoxides, and reactive aldehydes such as malondialdehyde (MDA), i.e., secondary products of lipid peroxidation [13]. In turn, MDA derivatives such as malondialdehyde acetaldehyde adducts may promote intra- or inter-molecular protein/DNA crosslinking, thus altering the biochemical properties of several proteins, including enzymes, carriers, and cytoskeletal, mitochondrial, and antioxidant proteins that are involved in aging and chronic diseases [14]. In patients with type 2 diabetes (T2D), elevated levels of MDA, a marker of lipid peroxidation, are associated with an increased risk of cardiovascular disease and complications of diabetes [15], thus contributing to the pathogenesis of atherosclerosis. An increase in lipid peroxidation levels has also been reported in the visceral adipose tissue (VAT) of postmenopausal women suffering from gynecological cancer [16]. 

Previous studies have demonstrated that oxidative stress is increased in obese patients, particularly in visceral obesity [9]. It has also been shown that postmenopausal women have greater levels of reactive oxygen species compared with premenopausal women [17,18] and men [19], indicating oxidative stress has been developed. It is likely that weight gain and the worsening of body composition play a role in increasing oxidative stress after menopause. However, very few data exist on the relationship between obesity and oxidative stress in postmenopausal women; specifically, there are no studies on this topic investigating the relationship between oxidative stress and adiposity, including with respect to VAT measured with a technique that provides high precision and accuracy such as dual-energy X-ray absorptiometry (DXA) [20]. The aim of this study was to fill this gap by investigating the association between oxidative stress and the key parameters related to fat accumulation. We show a positive correlation between BMI, VAT mass, and trunk fat percentage with oxidative stress markers, for which there is an almost two-fold increase in those markers among obese individuals compared to age-matched normal weight women. These findings underline the importance of keeping oxidative stress under control and managing fat-related parameters to reduce chronic cardiometabolic complications at such a critical age.

## 2. Materials and Methods

### 2.1. Study Subjects

We studied a subset of postmenopausal women who were participating in a previous cross-sectional study. Details of this subset of participants have been described previously [21]. Briefly, postmenopausal women (aged <65 years) undergoing hip arthroplasty for osteoarthritis were consecutively screened for participation in this study. Obesity was confirmed by the treating physician. Obesity was diagnosed when patients had a BMI ≥ 30 kg/m^2^, which is in accordance with the World Health Organization’s diagnostic criteria [22]. Malignancy was an exclusion criterion. Additionally, individuals treated with medications affecting bone such as estrogen, raloxifene, tamoxifen, bisphosphonates, teriparatide, denosumab, thiazolidinediones, glucocorticoids, anabolic steroids, and phenytoin, and those with hypercalcemia or hypocalcemia, hepatic or renal disorder, hypercortisolism, or currently consuming alcohol or taking anti-oxidant supplements, were excluded. This study was approved by the Ethics Committee of the Campus Bio-Medico University of Rome, and all participants provided written informed consent. 

### 2.2. Body Composition/Anthropometric Measurements

Body scans were conducted with a Lunar Prodigy^TM^ (GE Healthcare, Madison, WI, USA) DXA scanner. Body composition parameters were analyzed using the software enCORE^TM^ (version 17, 2016), and VAT was measured with the CoreScan^TM^ (GE Healthcare, USA). Quality control and calibration of the model were performed each day following the protocol provided by the manufacturer. All participants were examined after completing a fasting period of at least 8 h. Blood samples were collected 24 h before surgery.

For this analysis, the variables of interest derived from the DXA dataset were trunk fat mass (gr), trunk total mass (gr), and defined anatomical regions (android and gynoid) and visceral adipose tissue (VAT mass (g) and VAT volume, (cm^3^)). BMI was calculated as weight/height^2^ (kg/m^2^).

### 2.3. Serum Lipid Peroxidation (LPO) Assay

Serum LPO was analyzed by the thiobarbituric-acid-reactive substance (TBARS) assay (OxiSelect™ TBARS Assay kit, Cell Biolabs, Inc., San Diego, CA, USA) as previously reported [23]. Briefly, serum samples were immediately stored at −80 °C after blood sampling; then, 20 µL was incubated in sodium dodecylsulphate (SDS) lysis solution to denature the proteins. Then, thiobarbituric acid (TBA) was added, and the samples were incubated for 50 min at 95 °C. The tubes, before being centrifuged at 735× *g* for 15 min., were kept at room temperature. The supernatants were recovered, and optical density was measured by a multilabel plate reader at 490 nm (Victor X^TM^, Perkin Elmer, Inc., Waltham, MA, USA). 

### 2.4. Serum Levels of Total Hydroperoxide (TH)

Derivate-reactive oxygen metabolites (d-ROMs) Kit (Diacron Srl, Grosseto, Italy), was employed to measured TH levels, as reported in [24]. This method estimates the total amount of hydroperoxide present in a 10 μL serum sample by using a spectrophotometric procedure. Optical density was measured by a multilabel plate reader at 505 nm (Victor X^TM^, Perkin Elmer, USA), and the results were expressed in Carratelli units (UC) as conventional arbitrary units. The value of 1 Carratelli unit corresponds to a concentration of 0.08 mg/dL of hydrogen peroxide. A hydrogen peroxide calibration curve was developed using titrated H_2_O_2_ solutions.

### 2.5. Statistical Analysis

Data were analyzed using GraphPad Prism 9.0 (GraphPad Software, San Diego, CA, USA). Patients’ characteristics were described using means and standard deviation or counts and percentages. Group data are presented in boxplots with median and interquartile ranges; whiskers represent maximum and minimum values. We assessed data for normality. The unpaired *t*-test or the Chi square test were used to assess differences in the primary endpoints (d-ROMs) and in other variables between groups. Pearson’s correlation coefficients were used to assess the relationship between variables. Missing data were not imputed.

## 3. Results

### 3.1. Subject Characteristics

Of the 92 postmenopausal women originally enrolled in our previous study [21], 19 were excluded because they had diabetes, 11 because they were taking anti-oxidant nutritional supplements, and 31 because they did not have samples available for the oxidative stress analysis. A total of 31 postmenopausal women (12 with obesity and 19 of normal weight) were analyzed. As shown in Table 1, the clinical data of the study subjects highlighted that there were no differences in age and menopausal age between women with or without obesity. As expected, the BMI was significantly higher in the group with obesity compared to the group of women of normal weight. Fasting glucose levels were also higher in the subjects with obesity compared to those of normal weight (Table 1), although none of them were affected by diabetes as reported in their medical records. There were no differences in serum creatinine, urea, sodium, and potassium levels and the percentage of smokers between the groups (Table 1). The medications used by subjects with obesity included thyroid hormone replacement therapy (n = 1), statins (n = 1), and irbesartan/hydrochlorothiazide combination therapy (n = 1), whereas none of the normal weight subjects took any medications. 

### 3.2. Body Composition

Six subjects with obesity and seven subjects of normal weight underwent a DXA scan for an assessment of body composition. All the parameters related to body composition are reported in Table 2. Trunk total mass was significantly higher in the subjects with obesity compared to those without obesity, and so was trunk fat mass. Consistently, the percentage of trunk fat was higher in the subjects with obesity compared to those without obesity. Analysis of fat distribution highlighted that the subjects with obesity had both a higher percentage of android fat and gynoid fat compared to the normal weight subjects (Table 2). Finally, the levels of VAT mass and VAT volume were significantly greater in the subjects with obesity compared to the subjects of normal weight (Table 2).

### 3.3. Oxidative Stress Markers

We measured serum d-ROMs and LPO levels to study the levels of oxidative stress between the two groups (Figure 1). H_2_O_2_ levels were significantly higher in the subjects with obesity compared to those of normal weight (Figure 1A, *p* < 0.0001). The same trend was observed for MDA levels, which reflects the presence of lipid peroxidation damage in patients with obesity (Figure 1B, *p* < 0.0001).

Correlation analysis including all study subjects showed that both H_2_O_2_ (R = 0.7962, *p* < 0.0001) and MDA levels (R = 0.7749, *p* < 0.0001) increased with an increasing BMI (Figure 2A,B), but we found no correlation with fasting glucose levels for either H_2_O_2_ (R = 0.2369, *p* = 0.2159) or MDA (R = 0.2361, *p* = 0.2176) (Figure 2C,D).

Finally, oxidative stress markers showed a positive correlation (Figure 2E–H) with VAT mass (d-ROMs R = 0.7016, *p* = 0.0237; LPO R = 0.7892, *p* = 0.0066) and percentage of trunk fat (d-ROMs R = 0.7843, *p* = 0.0015: LPO R = 0.8983, *p* < 0.0001).

## 4. Discussion

In this cross-sectional study, total hydroperoxides and malondealdehyde, markers of oxidative stress and lipid peroxidation, respectively, were evaluated in postmenopausal women both without and with obesity. Our results suggest that oxidative stress is significantly increased in postmenopausal women living with obesity compared to their normal weight counterparts. Moreover, markers of oxidative stress were significantly associated with BMI, VAT, and trunk fat but not with fasting plasma glucose. To the best of our knowledge, this is the first study that specifically investigates the relationship between oxidative stress and DXA-derived measures of adiposity, including VAT, in postmenopausal women with or without obesity.

In fact, obesity is characterized by an excessive increase in fat storage, which, in turn, promotes lipid peroxidation. Studies suggest that this condition promotes oxidative stress by increasing NADPH oxidase activity (NOX) and decreasing mRNA expression and the activity of major antioxidant enzymes such as superoxide dismutase (SOD) and catalase (CAT). Ultimately, this pro-oxidant environment leads to a chronic proinflammatory state related to comorbidities and poor clinical outcome [25]. Oxidative stress is recognized as an important contributor to the pathogenesis of metabolic alterations including metabolic syndrome, type 2 diabetes and its complications [26], cancer [27], and CVD [12]. High total hydroperoxide plasma levels constitute an independent predictor of CVD events and mortality [28,29] and predict CAD in women [19]. Oxidative stress also plays a role in the decline of muscle mass and function (i.e., sarcopenia) observed during the aging process [10] and has been postulated to enhance bone resorption, possibly contributing to osteoporosis [30] and osteosarcopenic obesity [31].

The menopausal transition is associated with increases in total body fat mass and VAT [32]. It has been reported that in the shift from a premenopausal to postmenopausal status, VAT increases from 5–8% to 15–20% of total body fat [33,34]. In this study, we show that oxidative stress increases with an increasing VAT. This finding suggests that the increase in oxidative stress observed in postmenopausal women [17,19] might be due, at least in part, to changes in body composition after menopause. Our findings are consistent with most of the few previous studies that investigated the relationship between oxidative stress and obesity in postmenopausal women. In a study conducted in Mexico involving more than 500 pre- and postmenopausal women, oxidative stress markers were higher in overweight/obese women compared to normal weight women regardless of their menopausal status [35]. Furthermore, the authors found a weak but significant correlation (r = 0.298; *p* < 0.0001) between plasma MDA and the waist-to-height ratio, an indicator of central obesity. A positive correlation between oxidative stress (as assessed by a whole-blood free oxygen radical test) and abdominal obesity as assessed by either waist circumference (r = 0.345; *p* = 0.02) or the waist-to-hip ratio (r = 0.540; *p* = 0.0001) was reported by Cagnacci and colleagues [36]. A very limited number of studies assessed the relationship between DXA-derived measurements of central adiposity and oxidative stress in postmenopausal women. Pansini and colleagues found that antioxidant status (r = 0.46; *p* < 0.001) and hydroperoxide levels (r = 0.26; *p* < 0.05) increased with an increasing trunk fat mass [37] in pre- and postmenopausal women with a wide range of BMI values. In a different analysis of the same population, the same group reported a positive correlation between trunk fat mass and hydroperoxides and a negative correlation between trunk fat mass and antioxidant capacity (standardized multiple regression coefficients for the association of trunk fat mass with hydroperoxides and residual antioxidant power were equal to 0.324 and −0.495, respectively, for which *p* < 0.05 for both) among postmenopausal women aged ≤ 65 years without obesity [38]. In a similar population, Crist and colleagues showed that the androidal/gynoidal fat mass ratio (the sum of waist and hip fat mass divided by thigh fat mass) was a significant determinant of oxidative stress (estimated either by oxidized low-density lipoprotein or 15-isoprostane F2α) [39]. However, in a large study of older (≥50 years) adults followed-up for up to 14 years, longitudinal changes in hydroperoxide levels were significantly associated with a BMI ≥ 35 kg/m^2^, but there was no significant association with central obesity (either assessed by waist circumference or waist-to-hip ratio) in women [40]. Unlike previous studies, we included the DXA-derived measurement of VAT, which is strongly associated with cardiometabolic risk and is superior to anthropometric surrogates of abdominal obesity such as waist circumference or waist-to-hip ratio and waist-to-height-ratios [41]. Furthermore, DXA-derived VAT is highly correlated with VAT measured by magnetic resonance imaging and is preferable to the measurement of trunk fat, which includes subcutaneous adipose tissue [42]. 

The reduction in estrogen levels after menopause impacts body composition [3] and appears to induce molecular changes that may lead to an increase in oxidative stress in adipose tissue mainly due to reductions in antioxidant capacity. Data from animal studies indicate that the reduction in estrogen levels following an oophorectomy decreases the expression of glutathione peroxidase 3 (Gpx3), a gene involved in the protection of cells from ROS, in white adipose tissue [43]. There is also preclinical evidence that estrogen treatment reduces oxidative stress by promoting the expression of macrophage heme oxygenase-1, NAD(P)H:quinone oxidoreductase 1, and glutamate-cysteine ligase [16,44]. It has also been postulated that a decrease in antioxidant defenses might be mediated by low levels of follicle-stimulating hormone (FSH) in postmenopausal women. Klisic and colleagues showed that postmenopausal women were more likely to be overweight/obese and found a positive correlation between FSH levels and the activity of the antioxidant enzyme glutathione peroxidase. However, when multivariable regression analysis was performed, the relationship between FSH and glutathione peroxidase disappeared, and abdominal obesity, which was assessed by waist circumference, was the only variable significantly associated with lower FSH levels, suggesting that abdominal obesity may be a major determinant and modulator of antioxidant capacity as assessed by glutathione peroxidase [45]. Altogether, these results and our findings of a strong association between abdominal adiposity/VAT and markers of oxidative stress suggest that the increase in VAT due to estrogen reduction and aging [46,47] is a powerful mediator of oxidative stress in postmenopausal women. 

The lack of correlation between fasting plasma glucose and markers of oxidative stress is an unexpected finding that might be due to the relatively small range of glucose values measured glucose values. Furthermore, we did not measure plasma insulin concentrations and, therefore, were not able to calculate indices of insulin resistance, which have been shown to correlate with oxidative stress in several studies [48]. 

Our findings indicate the suitability of VAT accumulation as a therapeutic target to reduce oxidative stress (and, therefore, cardiometabolic risk) among postmenopausal women. Nutrition prevention strategies could help tackle VAT and VAT-accumulation-associated oxidative stress, as calorie restriction *per se* is associated with reductions in oxidative stress, due to an enhancement of mitochondrial biogenesis and turnover, antioxidant defenses, and decreased ROS production [49]. Several nutritional approaches have been proposed for postmenopausal women, which could help counteract the metabolic derangements associated with menopause [50].

The strengths of this study include its use of the d-ROMs test, i.e., a simple test that is already available as a point-of-care in vitro diagnostic and could, therefore, be easily implemented in clinical practice [29], and its use of DXA to determine VAT. Some limitations should be acknowledged. The number of participants was relatively low, and larger studies will be needed to confirm our results. It should be noted, however, that despite the small sample, the correlations between oxidative stress markers and adiposity parameters were very strong and significant. We did not include a control group of premenopausal women. Although this would have provided further insight into the relationship between oxidative stress and adiposity across different stages of a woman’s life, it has previously been shown that postmenopausal women have greater levels of oxidative stress compared with premenopausal women [17,18]. Therefore, we focused on the effect of obesity and fat accumulation, for which postmenopausal women of normal weight served as the control group. Additionally, we did not measure insulin levels or estrogen levels, as the focus of our study was the relationship between obesity, adiposity measures, and oxidative stress. Finally, this study’s cross-sectional design does not allow one to infer the direction of the associations found.

## 5. Conclusions

Menopause per se is associated with increased oxidative stress. We have shown that obesity and visceral fat are associated with a further increase in oxidative stress in postmenopausal women. These findings highlight the importance of keeping oxidative stress under control and preventing/managing excess adiposity to reduce chronic cardiometabolic complications at such a critical age. Clinicians who manage postmenopausal women should be aware of this abnormal association.

## Figures and Tables

**Figure 1 cells-12-01137-f001:**
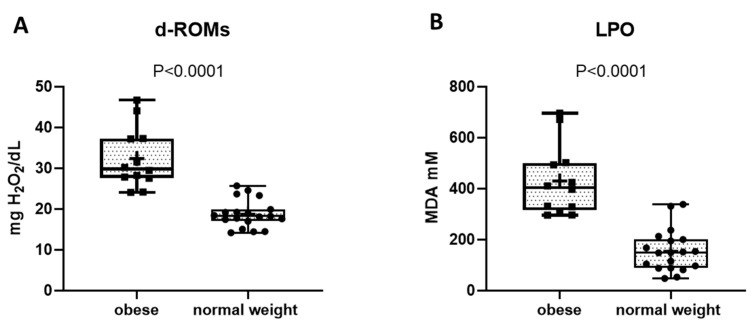
Oxidative stress and oxidative lipid damage in serum from study groups. (**A**) H_2_O_2_ and (**B**) MDA levels were evaluated in obese and normal weight postmenopausal women. All data represent the means of at least 3 replicates ± standard deviation. The analysis of difference between groups was performed using the unpaired *t*-test.

**Figure 2 cells-12-01137-f002:**
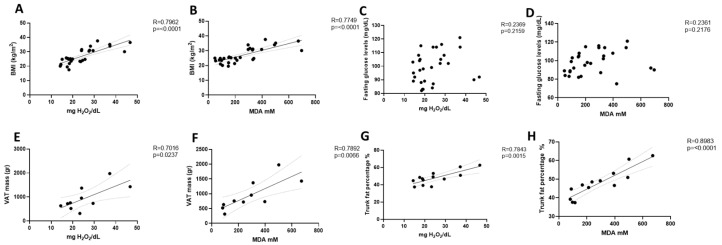
Correlation analysis of oxidative stress markers and body composition parameters. H_2_O_2_ and MDA levels were correlated with BMI (**A**,**B**), fasting glucose levels (**C**,**D**), VAT mass (**E**,**F**), and trunk fat percentage (**G**,**H**) in obese and normal weight postmenopausal women. Analysis was performed using Pearson comparison test. R indicates Pearson correlation coefficient.

**Table 1 cells-12-01137-t001:** Demographic and clinical features of the study subjects.

Parameters	Obesity (n = 12)	Normal Weight (n = 19)	*p* Value
Age (years)	71.0 (5.7)	67.7 (9.6)	0.2579
BMI (kg/m^2^)	32.4 (2.2)	23.1 (2.3)	<0.0001
Menopausal age (years)	49.7 (3.9)	49.5 (8.3)	0.9615
Fasting glucose (mg/dL)	105.4 (11.1)	96.3 (11.8)	0.0503
Creatinine (mg/dL)	0.76 (0.10)	0.71 (0.12)	0.3662
Urea (mg/dL)	39.3 (5.2)	39.7 (7.2)	0.9074
Sodium (mmol/L)	141.0 (1.06)	141.0 (2.14)	<0.9999
Potassium (mEq/L)	3.8 (0.2)	3.9 (0.3)	0.6706
Smokers (%)	3.00	12.50	0.0628

Data were analyzed using unpaired *t*-test and are presented as means (SD) or counts (percentage). Statistical significance was considered when *p* value was ≤0.05.

**Table 2 cells-12-01137-t002:** Body composition.

Parameters	Obesity (n = 6)	Normal Weight (n = 7)	*p* Value
Trunk fat mass (gr)	23,226 (4975)	13,845 (2963)	0.0012
Trunk total mass (gr)	42,895 (5309)	32,029 (3623)	0.0012
Trunk fat percentage (%)	53.83 (6.4)	42.83 (4.6)	0.0047
Android fat percentage (%)	56.57 (4.6)	44.77 (5.1)	0.0012
Gynoid fat percentage (%)	51.80 (4.4)	44.90 (3.8)	0.0140
VAT mass (gr)	1292 (479.3)	589.2 (178.9)	0.0159
VAT volume (cm^3^)	1397 (518.3)	636.8 (193.7)	0.0159

Data were analyzed using unpaired *t*-test and are presented as means (SD). VAT, visceral adipose tissue.

## Data Availability

The data presented in this study are available on request from the corresponding author.

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
