# Peer review of "Oxidative Stress in Postmenopausal Women with or without Obesity"

_cells, 2023, doi:10.3390/cells12081137_

Round 1

Reviewer 1 Report

In the current article, authors studied oxidative stress levels in postmenopausal women with or without obesity. Body composition was assessed by DXA and serum levels of lipid peroxidation and total hydroperoxides were measured by TBARS and d-ROMs assays, respectively. Authors found serum markers of oxidative stress levels were increased  in women with obesity compared to those with normal weight; malondialdehyde levels were increased in women with obesity. Correlation analysis showed that both markers of oxidative stress increased with increasing body mass index, visceral fat mass and trunk fat percentage, but not with fasting glucose levels. These observations are important and worth reporting. My only suggestion is change the word dangerous to abnormal in the conclusion sentence below

Clinicians who manage post-menopausal women should be aware of this dangerous association

Author Response

We thank the reviewer for the positive comment, we really value her/his opinion. We have changed the word dangerous with “abnormal” in the conclusions’ paragraph (Page 7, line 286).

Reviewer 2 Report

Please explain better the aim of your research given that the relation between obesity, menopause, age and oxidative stress is quite extensively studied? What is the novelty of your study?

In description of the groups, please detail the health condition of the subjects- diseases? medical treatment? laboratory investigations?

According to methods, oxidative stress evaluation was performed from plasma from patients subjects to arthroplasty but is not clear when the sample were obtained, before / after surgery?

Why you did not include a group of pre-menopausal patients?

Results should be more detailed and all abbreviations from figures should be explained (Ex. term “DROMs” has no explanation in the entire manuscript)

Author Response

  • Please explain better the aim of your research given that the relation between obesity, menopause, age and oxidative stress is quite extensively studied? What is the novelty of your study?

Authors’ response: We thank the reviewer for the opportunity to highlight the aim of the study and to better explain the relationship between oxidative stress, obesity and postmenopausal status. We have added the following sentences at the end of the introduction paragraph “It is likely that weight gain and worsening of body composition play a role in increasing oxidative stress after menopause. However, very few data exist on the relationship between obesity and oxidative stress in postmenopausal women, and specifically there are no study investigating the relationship between oxidative stress and adiposity in this setting, including VAT, measured with a technique that provides high precision and accuracy such as dual-energy X-ray absorptiometry (DXA). The aim of this study was to fill this gap, studying markers of oxidative stress in postmenopausal individuals and to investigate the association between oxidative stress and the key parameters related to fat accumulation. We show a positive correlation of BMI, VAT mass and trunk fat percentage with oxidative stress markers and almost doubling increase of those markers in individuals with obesity as compared to age-matched normal weight women. These findings underline the importance of keeping oxidative stress under control and manage fat related parameters to reduce chronic cardiometabolic complications at such critical age” (Page 2, lines 88-102).

  • In description of the groups, please detail the health condition of the subjects- diseases? medical treatment? laboratory investigations?

Authors’ response: We thank the reviewer for the opportunity to describe more in detail study subjects and clinical characteristics. All study subjects underwent a blood test 24 hours before hip surgery, as added in the method section (Page 3, line 126). In order to give more information about the health status, we added to table 1 the following clinical parameters: azotemia, sodium and potassium. We also added in the results section the description of these parameters (Page 4, line 146). Although fasting glucose levels were significantly higher in subject with obesity compared to normal weight, none of them were affected diabetes. We have specified this information in the description of subject characteristics (Page 4, lines 171-172). Finally, we added medical treatments in the same paragraph: “Medications used by subjects with obesity included thyroid hormone replacement therapy (n=1), statins (n=1) and irbesartan/hydrochlorothiazide combination therapy(n=1), while none of the normal weight subjects took any medications”. (Page 4, lines 175-177).

  • According to methods, oxidative stress evaluation was performed from plasma from patients subjects to arthroplasty but is not clear when the sample were obtained, before / after surgery?

Authors’ response: As reported in the previous response, we specified in the methods as follow: “Blood samples were collected 24 hours before surgery” (Page 3, line 126).

  • Why you did not include a group of pre-menopausal patients?

Authors’ response: We thank the reviewer for the constructive comment, and we agree that adding a group of pre-menopausal women would have been of interest. As it has previously been shown that postmenopausal women have greater levels of oxidative stress as compared with premenopausal women [Sanchez-Rodriguez et al, 2012, doi:10.1097/gme.0b013e318229977d; Bourgonje et al, 2020, doi:10.3390/ijms21010314], we focused on the effect of obesity and fat accumulation in postmenopausal women, therefore postmenopausal women with normal weight served as the control group. We have added this clarification at the end of the discussion (page 8, lines 321-326yy).

  • Results should be more detailed and all abbreviations from figures should be explained (Ex. term “DROMs” has no explanation in the entire manuscript)

Authors’ response:

As suggested by the reviewer we have specified the d-ROMs abbreviation in the description of the methodology “derivates reactive oxygen metabolites” (Page 3, line 144). LPO abbreviation was already explained in the methods section “Serum lipid peroxidation assay” (Page 3, line 132).

Reviewer 3 Report

This study compared oxidative stress markers in postmenopausal women with and without obesity. Although it is an interesting study, it has the following shortcomings.

Major Comments

1.      Menstruation and menopause are often mentioned in the introduction and discussion of this study. The issue of obesity is unclear.

2.      The drawback of this study is the small number of subjects for whom the final results were obtained. In other words, there is concern about coincidence. It is necessary to increase the number of subjects or calculate the sample size to show the validity of this study.

Minor Comments

P2 L92

Subjects

  It shows the flow chart so that you can see the exclusion of the target and the final analysis target. Add an explanation why the number of subjects is reduced in the resulting Body composition.

P3 L122

Statistical analysis

Please describe a sample size calculation. Please refer to the author's previous papers or other papers for sample size calculation.

Why did the author use the non-parametric Mann-Whitney U test for comparison of group data, and the parametric Pearson's correlation for correlation? A normality test is required when using Pearson's correlation.

P3 L137

Table

Align the significant digits (number of digits) of the p value.

Please enter 4 digits after the decimal point.

In the footnote of the table, please indicate what statistic was used to obtain the p value.

Table 2

In the footnote of the table, please indicate what statistic was used to obtain the p value.

Figure 2

Lines in the figure that do not show a significant association are not required. Also, since the width of the vertical axis differs in the figure, the author should align it.

In the H graph, there are plots where the MDA is around 0. However, in the graph of D, there is no plot where MDA is near 0. Are the subjects A to D different from the subjects E to H?

Didn't some of the subjects A to D undergo tests E to H?

Discussion

P5 L182

Didn't this study include patients with type 2 diabetes? If not included, the author should provide Table 1 with blood glucose levels etc.

P5 L190

Discussion.

P6 L199L216

It is an enumeration of previous research, and the focus of the discussion of this study is unclear. The results of previous studies are also important. However, the physiological mechanism by which the results of this study were obtained should be considered in discussion.

P6 L242

The link between FSH and VAT is unclear.

P6 limitation

Although there is no significant difference in smoking prevalence, the tables for smoking prevalence are quite different. The possibility that this difference in smoking prevalence affects oxidative stress markers cannot be denied.

P7 L272

[possibly increasing cardiometabolic and cancer risk] cannot be definitively demonstrated in this study. This study shows that obesity increases oxidative stress markers.

Author Response

This study compared oxidative stress markers in postmenopausal women with and without obesity. Although it is an interesting study, it has the following shortcomings.

Major Comments

  1. Menstruation and menopause are often mentioned in the introduction and discussion of this study. The issue of obesity is unclear.

      Authors’ response: We apologize for the lack of clarity. In the introduction, we have better clarified that weight gain and worsening of body composition occur after menopause, leading to overweight/obesity and excess morbidity and mortality. We also specify that it is likely that weight gain and worsening of body composition play a role in the increase in oxidative stress after menopause that has been reported by other authors. This laid the ground for our study (Page 2, lines 52-55).  

  1. The drawback of this study is the small number of subjects for whom the final results were obtained. In other words, there is concern about coincidence. It is necessary to increase the number of subjects or calculate the sample size to show the validity of this study.

      Authors’ response: Our primary endpoint was the difference in oxidative stress markers (d-ROMs). As per the Reviewer’s request, we performed a post-hoc power calculation to assess the validity of the study.  Post hoc analysis demonstrated that effect size was 2.35, with a power of 0.99, confirming that sample size was sufficient to achieve adequate statistical power.

Minor Comments

P2 L92

Subjects

It shows the flow chart so that you can see the exclusion of the target and the final analysis target. Add an explanation why the number of subjects is reduced in the resulting Body composition.

Authors’ response: We really thank the reviewer for this suggestion. However, considering the relatively small number of enrolled patients for the study, we added this information in the description of the subject characteristics. “Of the 92 postmenopausal women originally enrolled in our previous study, 19 were excluded because they had diabetes, 11 were taking antioxidant nutritional supplement and 31 did not have samples available for the oxidative stress analysis”.  (Page 4, line 163-165)

P3 L122

Statistical analysis

Please describe a sample size calculation. Please refer to the author's previous papers or other papers for sample size calculation.

Why did the author use the non-parametric Mann-Whitney U test for comparison of group data, and the parametric Pearson's correlation for correlation? A normality test is required when using Pearson's correlation.

      Authors’ response: We apology for the oversight in the description of the statistical analysis and we thank the reviewer for the careful reading. As per the Reviewer’s request, we performed a post-hoc power calculation to assess the validity of the study.  Post hoc analysis demonstrated that effect size was 2.35, with a power of 0.99, confirming that sample size was sufficient to achieve adequate statistical power. We assessed our data for normality, which confirmed normal distribution. Accordingly, we used Pearsons’ correlation to look at association between oxidative stress levels and body composition parameters and unpaired t-test to calculate differences between groups. We modified the name of the statistical test in the corresponding paragraph (Pag. 4, lines 156-160).

P3 L137

Table1

Align the significant digits (number of digits) of the p value.

Please enter 4 digits after the decimal point. In the footnote of the table, please indicate what statistic was used to obtain the p value.

Authors’ response: We modified table 1 according to your suggestions.

Table 2

In the footnote of the table, please indicate what statistic was used to obtain the p value.

Authors’ response: We modified the footnotes according to your suggestions.

Figure 2

Lines in the figure that do not show a significant association are not required. Also, since the width of the vertical axis differs in the figure, the author should align it.

In the H graph, there are plots where the MDA is around 0. However, in the graph of D, there is no plot where MDA is near 0. Are the subjects A to D different from the subjects E to H?

Didn't some of the subjects A to D undergo tests E to H?

Authors’ response: We apologize for the mistake in figure H, which was caused by a typo in one MDA value. We corrected the corresponding r coefficient in the manuscript (Page 6, line 212) and we also changed the graph of figure H. Some values were missing, therefore the number of subjects slightly differs across figures. We acknowledge in the statistic paragraph that missing values were not imputed.

Discussion

P5 L182

Didn't this study include patients with type 2 diabetes? If not included, the author should provide Table 1 with blood glucose levels etc.

Authors’ response:

We thank the reviewer for the opportunity to describe more in detail study subjects and clinical characteristics. All study subjects underwent a blood test 24 hours before hip surgery, as added in the method section (Page 3, line 26). In order to give more information about the health status, we added to table 1 the following clinical parameters: azotemia, sodium and potassium. We also added in the results section the description of these parameters (Page 4, lines 173). Although fasting glucose levels were significantly higher in subject with obesity compared to normal weight, none of them were affected diabetes. We have specified this information in the description of subject characteristics (Page 4, lines 171-172). Finally, we added medical treatments in the same paragraph: “Medications used by subjects with obesity included thyroid hormone replacement therapy (n=1), statins (n=1) and irbesartan/hydrochlorothiazide combination therapy(n=1), while none of the normal weight subjects took any medications”. (Page 4, lines 175-177).

P5 L190

Discussion.

P6 L199~L216

It is an enumeration of previous research, and the focus of the discussion of this study is unclear. The results of previous studies are also important. However, the physiological mechanism by which the results of this study were obtained should be considered in discussion.

Authors’ response: We agree with the reviewer that is of interest to improve the discussion by better explaining physiological mechanism by which oxidative stress is increased in postmenopausal “Obesity is characterized by an excessive increase in fat storage, which in turn promotes lipid peroxidation.  This condition promotes oxidative stress by increasing NADPH oxidase activity (NOX) and decreasing mRNA expression and activities of major antioxidant enzymes such as, superoxide dismutase (SOD) and catalase (CAT). This prooxidant environment ultimately leads to chronic proinflammatory state related to comorbidities and poor clinical outcome (Evans J, 2002).”

P6 L242

The link between FSH and VAT is unclear.

Authors’ response:

We apologize for the lack of clarity. A previous study (Klisic and coll.) reported that low FSH was associated with reduced antioxidant capacity, but when multivariable regression analysis was performed, the relationship between FSH and glutathione peroxidase disappeared and abdominal obesity, as assessed by waist circumference, was the only variable significantly associated with lower FSH levels, suggesting that FSH might be a mediator and abdominal obesity a major determinant and modulator of antioxidant capacity as assessed by glutathione peroxidase. This has been clarified in the text (page 7, lines 293-297). Also, please note that only the relationship between FSH and abdominal adiposity as assessed by waist circumference was mentioned. Although abdominal adiposity is a surrogate biomarker of visceral adipose tissue, this was not measured directly and therefore is not mentioned in the discussion of the study by Klisic and coll. Accordingly, we specified that “these and our findings of a strong association between abdominal adiposity/VAT and markers of oxidative stress suggest that the increase in VAT secondary to estrogen reduction and aging is a powerful mediator of oxidative stress in postmenopausal women”, and provide two references to support the statement that VAT increases with estrogen reduction and aging (Bjune 2022 and Ko 2021).

P6 limitation

Although there is no significant difference in smoking prevalence, the tables for smoking prevalence are quite different. The possibility that this difference in smoking prevalence affects oxidative stress markers cannot be denied.

Authors’ response: We agree with the reviewer. In the discussion we acknowledge that “The study was adequately powered to detect differences between groups in the primary endpoint, but we can not exclude that the study was underpowered to detect significant differences in other variables such as the proportion of smokers, which was numerically higher in the subjects with obesity”.

P7 L272

[possibly increasing cardiometabolic and cancer risk] cannot be definitively demonstrated in this study. This study shows that obesity increases oxidative stress markers.

Authors’ response: We agree with the reviewer and we changed the sentence as follows: “These findings highlight the importance of keeping oxidative stress under control and prevent/manage excess adiposity to reduce chronic cardiometabolic complications at such critical age” (Pag. 8, lines 334-336).

Round 2

Reviewer 2 Report

The authors performed required changes.

Recommendation: publish.